# Research on Nonlinear Compensation of the MEMS Gyroscope under Tiny Angular Velocity

**DOI:** 10.3390/s22176577

**Published:** 2022-08-31

**Authors:** Chunhua Ren, Dongning Guo, Lu Zhang, Tianhe Wang

**Affiliations:** The Key Laboratory of Optoelectronic Technology and System, Ministry of Education, Chongqing University, Chongqing 400030, China

**Keywords:** MEMS gyroscope, nonlinearity, tiny angular velocity, steepest descent method, Fourier series

## Abstract

The Micro-Electro-Mechanical System (MEMS) gyroscope has been widely used in various fields, but the output of the MEMS gyroscope has strong nonlinearity, especially in the range of tiny angular velocity. This paper proposes an adaptive Fourier series compensation method (AFCM) based on the steepest descent method and Fourier series residual correction. The proposed method improves the Fourier series fitting method according to the output characteristics of the MEMS gyroscope under tiny angular velocity. Then, the optimal weights are solved by the steepest descent method, and finally the fitting residuals are corrected by Fourier series to further improve the compensation accuracy. In order to verify the effectiveness of the proposed method, the angle velocity component of the earth’s rotation is used as the input of the MEMS gyroscope to obtain the output of the MEMS gyroscope under tiny angular velocities. Experimental characterization resulted in an input angular velocity between −0.0036°/s and 0.0036°/s, compared with the original data, the polynomial compensation method, and the Fourier series compensation method, and the output nonlinearity of the MEMS gyroscope was reduced from 1150.87 ppm, 641.13 ppm, and 250.55 ppm to 68.89 ppm after AFCM compensation, respectively, which verifies the effectiveness and superiority of the proposed method.

## 1. Introduction

The Micro-Electro-Mechanical System (MEMS) gyroscope is a critical component of inertial navigation systems (INS) [1]. The advantages of the MEMS gyroscope include its low cost, compact volume, low power consumption, and easy integration. As a result, the MEMS gyroscope is widely used in oil drilling and geological research [2,3]. However, the MEMS gyroscope’s precision is affected by various errors [4].

The error of the MEMS gyroscope consists of random error and deterministic error [5]. Random errors are caused by uncertain factors and have no obvious repeatability. It can be analyzed by statistics and filtering methods, such as the Auto Regression Moving Average (ARMA) model, the Adaptive Kalman Filter (AKF), and Allan variance [6,7,8]. Deterministic errors include misalignment, zero bias, and nonlinearity [9,10]. The compensation of the MEMS gyroscope misalignment and the zero bias have been extensively reported [11,12,13,14]. However, the nonlinearity of the MEMS gyroscope is affected by geometric and material effects, electrostatic actuation, capacitance detection, and other factors. The source of the error is complex, which results in compensation difficulties [15].

In earlier work, the nonlinear compensation method of the MEMS gyroscope has been studied, which can be mainly divided into the following methods:The traditional compensation method: Refers to the IEEE standard format guide [16,17], where the relationship between the input and output angular velocity of the MEMS gyroscope is established, and the optimal relationship is found by polynomial fitting [18,19,20], so as to realize the nonlinear error compensation. However, the performance of polynomial fitting depends on the accuracy of the compensation model and the repeatability of the system output, which is not universal.Virtual Coriolis force-based nonlinear compensation method: With a specific resonator structure (force rebalance comb), the MEMS gyroscope outputs an equivalent angular velocity signal by applying an electrical excitation signal and then compensates for the nonlinearity of the MEMS gyroscope according to the output [21,22]. However, the additional vibration introduced by the virtual Coriolis force method, which is coupled to the gyroscope sensitive end, results in additional nonlinear errors.Artificial intelligence algorithm: The output model of the MEMS gyroscope was established by using fuzzy logic [23] and neural networks [24]. When the input angular velocity is −60°/s to 60°/s and the interval is 3°/s, the nonlinear error of the MEMS gyroscope is about 140 ppm. However, artificial intelligence algorithms require a large number of samples to train.In-run compensation method: The sources of MEMS gyroscope nonlinear error were investigated, and a nonlinear error correction method that does not require system calibration or data fitting was proposed, which can be applied to resonant gyroscopes in amplitude-modulated (AM) mode in general [25,26]. When the input angular velocity is ±0.1°/s, ±0.2°/s, ±0.5°/s, ±1°/s, and ±2°/s, the compensation method is verified.

These studies have effectively reduced the nonlinear error of the MEMS gyroscope at high angular velocity, but the compensation of MEMS gyroscope output nonlinearity (nonlinear behavior, or deviation from linear dependence) under tiny angular velocity (from −0.0036°/s to 0.0036°/s) has not been studied. Due to the strong nonlinearity of MEMS gyroscope output, it is difficult for the MEMS gyroscope to accurately measure the earth’s rotation angle velocity component. Therefore, the MEMS gyroscope is rarely used in north-seeking. With the increasing demand for MEMS gyroscope north-seeking accuracy, more and more fields need to use the MEMS gyroscope for north-seeking. The nonlinear compensation of the MEMS gyroscope output is particularly important, especially in the case of tiny angular velocity (angular velocity of earth’s rotation). Therefore, this paper studies the nonlinear error compensation method of the MEMS gyroscope under tiny angular velocity. An adaptive Fourier series compensation method (AFCM) based on the steepest descent method and Fourier series residual correction is proposed. AFCM achieves a better compensation effect by improving the Fourier series fitting method. It then iteratively solves the optimal weights using the steepest descent method and further improves the compensation accuracy by correcting the residuals of the fitted model with the Fourier series. The experimental results show that the proposed method can effectively reduce the nonlinear error of the MEMS gyroscope under tiny angular velocity.

The paper is organized as follows. Section 2 introduces the nonlinear mechanism of the MEMS gyroscope and establishes the output model of the MEMS gyroscope at tiny angular velocities. In Section 3, the method of nonlinear compensation is reviewed. Section 4 represents the experimental process and results. Section 5 is the concluding remarks and future work.

## 2. The Output Model of the MEMS Gyroscope under Tiny Angular Velocity

### 2.1. Nonlinear Analysis of the MEMS Gyroscope

Polysilicon, or crystalline silicon, is the main material of the MEMS gyroscope, which is fabricated by microfabrication. The MEMS gyroscope is a micron-scale device, so it has the characteristics of micro-size semiconductor processing, which leads to the weak signals being vulnerable to interference and complex error sources, limiting the accuracy of the device. In the machining process, the machining accuracy of device size is difficult to control, which affects the performance of the MEMS gyroscope. For the MEMS gyroscope, noise is an important factor affecting the performance of the gyroscope. Noise is a kind of random signal, which will affect the useful signal measured by the system. When the signal-to-noise ratio of the system is too low, the useful signal will be seriously disturbed, and even submerged by noise, so the noise largely determines the accuracy of the MEMS gyroscope output. The main noise sources of the MEMS gyroscope include thermal noise [27,28] and flicker noise (1/f noise) [29]. Thermal noise includes mechanical thermal noise and electronic thermal noise, which is caused by the random thermal motion of molecules or electrons. The random fluctuation of the conductance of the conductor contact point causes 1/f noise. The relationship between signal, noise, and the MEMS gyroscope is shown in Figure 1. In addition, the nonlinearity of capacitance detection also leads to the nonlinearity of MEMS gyroscope output [30,31].

The structure of the MEMS gyroscope is shown in Figure 2, which consists of a vibratory mass, driving electrodes, sensing electrodes, two anchors, supporting beams, an inner frame, and an outer frame [32]. When there is angle movement on the MEMS gyroscope’s input axis, because of the Coriolis effect, the vibratory mass will twist and a Coriolis inertia torque ($M_{c}$) is generated, resulting in the change of the capacitance value of the sensing electrode. Torsion amplitude ($\theta_{k^{\prime }}$) in the output axis is sensed by the sensing electrodes, and the capacitance difference (ΔC) between the two sensing electrodes is output. Through the measurement of the capacitance change, the input angular velocity ($\tilde{\Omega }$) can be indirectly calculated.

The $M_{c}$ can be expressed as:(1)$$M_{c}=\Theta_{J}{\ddot{\theta }}_{N}\tilde{\Omega }$$
where $\Theta_{J}$ is the equivalent moment of inertia of the inner frame and ${\ddot{\theta }}_{N}$ is the constant amplitude vibration of the inner frame.

In accordance with the Laplace transform principle, the movement of the outer frame of the MEMS gyroscope subjected to $M_{c}$ is described by the equation:(2)$$\begin{array}{cc} \theta_{k^{\prime }} & =\frac{M_{c}}{J_{k^{\prime }}S^{2}+D_{k^{\prime }}S+B_{k^{\prime }}}=\frac{\Theta_{J}S{\ddot{\theta }}_{N}}{J_{k^{\prime }}(S^{2}+2\psi_{k^{\prime }}{ℵ}_{k^{\prime }}S+{ℵ}_{k^{\prime }}^{2})}\cdot \tilde{\Omega }\\ & =\frac{\Theta_{J}{ℵ}_{D}{\ddot{\theta }}_{N}∠90}{J_{k^{\prime }}{ℵ}_{k^{\prime }}^{2}\sqrt{{\left[1-{\left(\frac{{ℵ}_{D}}{{ℵ}_{k^{\prime }}}\right)}^{2}\right]}^{2}+{\left(\frac{2\psi_{k^{\prime }}{ℵ}_{D}}{{ℵ}_{k^{\prime }}}\right)}^{2}}}\cdot \tilde{\Omega } \end{array}$$
where $J_{k^{\prime }}$ is the moment of inertia in $k^{\prime }$ axis, $k^{\prime }=x,y,z$, $\theta_{k^{\prime }}$ is torsion amplitude in output axis, S is complex frequency, S=σ+jω, $D_{k^{\prime }}$ and $\psi_{k^{\prime }}$ are the damping coefficient and ratio, $\psi_{k^{\prime }}=D_{k^{\prime }}/2\sqrt{B_{k^{\prime }}\cdot J_{k^{\prime }}}$, ${ℵ}_{D}$ and ${ℵ}_{k^{\prime }}$ are the resonant radial frequency of drive and output axis, and $B_{k^{\prime }}$ is the rigidity of output axis, $B_{k^{\prime }}={ℵ}_{k^{\prime }}^{2}J_{k^{\prime }}$.

The capacitance difference of the MEMS gyroscope, as a function of torsion displacement, $\theta_{k^{\prime }}$, is given as:(3)$$\begin{array}{cc} \Delta C & =f(\theta_{k^{\prime }})=f\left(\frac{\Theta_{J}{ℵ}_{D}{\ddot{\theta }}_{N}∠90}{J_{k^{\prime }}{ℵ}_{k^{\prime }}^{2}\sqrt{{\left[1-{\left(\frac{{ℵ}_{D}}{{ℵ}_{k^{\prime }}}\right)}^{2}\right]}^{2}+{\left(\frac{2\psi_{k^{\prime }}{ℵ}_{D}}{{ℵ}_{k^{\prime }}}\right)}^{2}}}\cdot \tilde{\Omega }\right)\\ & =-\frac{\kappa l_{s}}{\theta_{k^{\prime }}}\ln\frac{h_{0}^{2}-{\left(s+b\right)}^{2}\theta_{k^{\prime }}^{2}}{h_{0}^{2}-{\left(s\cdot \theta_{k^{\prime }}\right)}^{2}} \end{array}$$
where κ is the dielectric constant between plates, $l_{s}$ is the effective length of the sensitive electrode, $h_{0}$ is the initial distance between the drive plate and the inner frame, s is the distance between the sensitive electrode, and the width of the plate is denoted by b.

It can be seen from Equation (3) that ΔC is not a linear function of $\theta_{k^{\prime }}$, that is, the output of the MEMS gyroscope has nonlinear error. Through expansion of natural logarithm in Equation (3) by Taylor series and omitting higher order terms, we can obtain:(4)$$\Delta C\approx P(\theta_{k^{\prime }})=\frac{\kappa A(2s+b)}{h_{0}^{2}-{\left(s\cdot \theta_{k^{\prime }}\right)}^{2}}\theta_{k^{\prime }}$$

In the formula, A is the effective area of the sensitive electrode.

From the above equation, the nonlinear error of the MEMS gyroscope can be expressed as:(5)$$\begin{array}{cc} \Delta {ℜ}_{c} & =f(\theta_{k^{\prime }})-P(\theta_{k^{\prime }})\\ & =\frac{A(2s+b)\theta_{k^{\prime }}^{2}-l_{s}[h_{0}^{2}-{\left(s\cdot \theta_{k^{\prime }}\right)}^{2}]\ln\frac{h_{0}^{2}-{\left(s+b\right)}^{2}\theta_{k^{\prime }}^{2}}{h_{0}^{2}-{\left(s\cdot \theta_{k^{\prime }}\right)}^{2}}}{\theta_{k^{\prime }}h_{0}^{2}-s^{2}\cdot \theta_{k^{\prime }}^{3}}\kappa \end{array}$$

It can be seen from Equation (5) and Figure 1 that the nonlinearity of MEMS gyroscope output is affected by the nonlinearity of capacitance detection, and in the low input speed range, the effective signal is small, the machining error and noise are prominently affected, and the MEMS gyroscope output error nonlinearity is more significant.

### 2.2. Establishment of MEMS Gyroscope Output Model

To compensate for the nonlinearity of the MEMS gyroscope in a tiny angular velocity range and improve its output precision, it is necessary to establish the output model of the MEMS gyroscope mathematically. The mathematical model expresses the relationship between the input and output of the MEMS gyroscope [33]. Therefore, the gyroscope’s output model can be expressed as follows:(6)$$G=\tilde{K}\omega_{i}+E^{\prime }\omega_{j}+\tilde{D}A+G_{0}+\varepsilon$$
where G is the output angular velocity of the MEMS gyroscope, ω denotes the true angular velocity, and A is the acceleration. They are shown as follows:(7)$$\omega ={\left[\begin{array}{ccc} \omega_{x} & \omega_{y} & \omega_{z} \end{array}\right]}^{T}$$
(8)$$A={\left[\begin{array}{ccc} A_{x} & A_{y} & A_{z} \end{array}\right]}^{T}$$

Gyroscope zero bias errors and random errors are defined as:(9)$$G_{0}={\left[\begin{array}{ccc} G_{0x} & G_{0y} & G_{0y} \end{array}\right]}^{T}$$
(10)$$\varepsilon ={\left[\begin{array}{ccc} \varepsilon_{x} & \varepsilon_{y} & \varepsilon_{z} \end{array}\right]}^{T}$$

$\tilde{K}$ is scale factor of the MEMS gyroscope. It is shown as follows:(11)$$\tilde{K}=\left[\begin{array}{ccc} {\tilde{K}}_{x} & 0 & 0\\ 0 & {\tilde{K}}_{y} & 0\\ 0 & 0 & {\tilde{K}}_{z} \end{array}\right]$$

$E^{\prime }$ is installation error of the MEMS gyroscope. $\tilde{D}$ is bias error related to acceleration. They are shown as follows:(12)$$E^{\prime }=\left[\begin{array}{ccc} 0 & {E^{\prime }}_{xy} & {E^{\prime }}_{xz}\\ {E^{\prime }}_{yx} & 0 & {E^{\prime }}_{yz}\\ {E^{\prime }}_{zx} & {E^{\prime }}_{zy} & 0 \end{array}\right]$$
(13)$$\tilde{D}=\left[\begin{array}{ccc} {\tilde{D}}_{xx} & {\tilde{D}}_{xy} & {\tilde{D}}_{xz}\\ {\tilde{D}}_{yx} & {\tilde{D}}_{yy} & {\tilde{D}}_{yz}\\ {\tilde{D}}_{zx} & {\tilde{D}}_{zy} & {\tilde{D}}_{zz} \end{array}\right]$$
where ${E^{\prime }}_{ij}$ is the misalignment caused by the non-orthogonality axes of the gyroscope. The misalignment ${\tilde{D}}_{ij}$ denotes that the acceleration sensed by the i-axis accelerometer is caused by the input acceleration of the j-axis accelerometer.

Inserting Equations (7)–(13) into Equation (6), we obtain:(14)$$\left[\begin{array}{c} G_{x}\\ G_{y}\\ G_{z} \end{array}\right]=\left[\begin{array}{ccc} {\tilde{K}}_{x} & {E^{\prime }}_{xy} & {E^{\prime }}_{xz}\\ {E^{\prime }}_{yx} & {\tilde{K}}_{y} & {E^{\prime }}_{yz}\\ {E^{\prime }}_{zx} & {E^{\prime }}_{zy} & {\tilde{K}}_{z} \end{array}\right]\left[\begin{array}{c} \omega_{x}\\ \omega_{y}\\ \omega_{z} \end{array}\right]+\left[\begin{array}{ccc} {\tilde{D}}_{xx} & {\tilde{D}}_{xy} & {\tilde{D}}_{xz}\\ {\tilde{D}}_{yx} & {\tilde{D}}_{yy} & {\tilde{D}}_{yz}\\ {\tilde{D}}_{zx} & {\tilde{D}}_{zy} & {\tilde{D}}_{zz} \end{array}\right]\left[\begin{array}{c} A_{x}\\ A_{y}\\ A_{z} \end{array}\right]+\left[\begin{array}{c} G_{0x}\\ G_{0y}\\ G_{0z} \end{array}\right]+\left[\begin{array}{c} \varepsilon_{x}\\ \varepsilon_{y}\\ \varepsilon_{z} \end{array}\right]$$

To obtain the output of the MEMS gyroscope at a tiny angular velocity, the earth’s rotation angular velocity component is used as the input of the gyroscope. As shown in Figure 3, the gyroscope output appears to be a cosine signal when the angle between the gyroscope sensitive axis and the Earth’s rotation axis varies.

It can be seen from Figure 3 that the input of the MEMS gyroscope and accelerometer is [34]:(15)$$\left\{ \begin{array}{l} \omega_{x}=\omega_{ie}\cos L\sin\alpha \\ \omega_{y}=\omega_{ie}\cos L\cos\alpha \\ \omega_{z}=\omega_{ie}\sin L\\ A_{x}=0\\ A_{y}=0\\ A_{z}=-g \end{array}\right.$$
where $\omega_{ie}$ represents the earth’s angular velocity, L denotes the earth’s latitude, and α indicates the angle between the sensitive axis of the MEMS gyroscope and true north.

Substituting Equation (15) into Equation (14), the output model of the MEMS gyroscope under tiny angular velocity is shown as:(16)$$\{\begin{array}{l} G_{x}={\tilde{K}}_{x}\omega_{ie}\cos L\sin\alpha +{E^{\prime }}_{xy}\omega_{ie}\cos L\cos\alpha +{E^{\prime }}_{xz}\omega_{ie}\sin L-{\tilde{D}}_{xz}g+G_{0x}+\varepsilon_{x}\\ G_{y}={\tilde{K}}_{y}\omega_{ie}\cos L\cos\alpha +{E^{\prime }}_{yx}\omega_{ie}\cos L\sin\alpha +{E^{\prime }}_{yz}\omega_{ie}\sin L-{\tilde{D}}_{yz}g+G_{0y}+\varepsilon_{y}\\ G_{z}={\tilde{K}}_{z}\omega_{ie}\sin L+{E^{\prime }}_{zx}\omega_{ie}\cos L\cos\alpha +{E^{\prime }}_{zy}\omega_{ie}\cos L\sin\alpha -{\tilde{D}}_{zz}g+G_{0z}+\varepsilon_{z} \end{array}$$

The output model of the MEMS gyroscope includes installation error, acceleration bias error, zero bias error, and random error. These errors can be calibrated by the rate method and the position method. After calibrating the above errors, the linearity of the MEMS gyroscope is better at large angular velocity, but at tiny angular velocity, the linearity becomes poor. Therefore, this paper focuses on the nonlinear compensation of MEMS gyroscope output at tiny angular velocity.

## 3. Compensation Method

### 3.1. The Traditional Polynomial Compensation Method

The classical strategy to compensate for MEMS gyroscope nonlinearity is polynomial fitting. The regression equation can be obtained [35,36]:(17)$$\hat{\Omega }(\vec{G})=a_{0}+a_{1}\vec{G}+a_{2}{\vec{G}}^{2}+\cdots +a_{n}{\vec{G}}^{n}$$
where $\vec{G}$ is the MEMS gyroscope output angular velocity, $\hat{\Omega }(\vec{G})$ is the fitting angular velocity, and $a_{0},a_{1},\cdots ,a_{n}$ is the parameter of the established model.

The difference between the value calculated by the regression equation and the desired value is as follows:(18)$$I=\sum_{i=0}^{m}{\left[\hat{\Omega }(\vec{G})-\omega_{i}\right]}^{2}=\sum_{i=0}^{m}(\sum_{h=0}^{n}a_{h}{\vec{G}}_{i}^{h}-\omega_{i}{)}^{2}$$
where $\omega_{i}$ is the ideal output value of the MEMS gyroscope.

According to the least mean squares theory, the first-order partial derivative of I should be set to the minimum to obtain the optimum value for the coefficients:(19)$$\partial I/\partial a_{j}=2\sum_{i=0}^{m}(\sum_{h=0}^{n}a_{h}{\vec{G}}_{i}^{h}-\omega_{i}){\vec{G}}_{i}^{j}\to \min$$

The linear equations about $a_{0},a_{1},\cdots ,a_{n}$ can be obtained. Thus, the regression equation with order n can be obtained.

### 3.2. The Adaptive Fourier Series Compensation Method (AFCM)

There are many factors that cause the nonlinearity of the MEMS gyroscope. The polynomial compensation method is difficult to accurately describe this characteristic, and the calculation amount is also large [37]. Therefore, this paper proposes an adaptive Fourier series compensation method (AFCM). It can be seen from the output model of the tiny angular velocity of the MEMS gyroscope that the model includes not only sine and cosine terms but also some additional terms. The Fourier series fitting method is improved in order to better fit the output of the MEMS gyroscope at tiny angular velocity. Then, the optimal weight is iteratively obtained by the steepest descent method. Finally, the Fourier series is used to eliminate the residuals of the fitting model, so as to achieve the best fitting effect.

A very high-precision-rate turntable is needed in order to achieve the input of tiny angular velocity. Therefore, this paper adopts the multi-position turntable experiment in Section 2.2. It can be seen from Equation (16) and Figure 3 that the output of the MEMS gyroscope can be approximated as a periodic trigonometric function satisfying the Dirichlet condition. As a result, the output of the MEMS gyroscope can be decomposed into a Fourier series, and its trigonometric function can be expressed as:(20)$$x(t)={a^{\prime }}_{0}+\sum_{n=1}^{\infty }{a^{\prime }}_{n}\cdot \cos(n\tilde{w}t)+\sum_{n=1}^{\infty }{b^{\prime }}_{n}\cdot \sin(n\tilde{w}t)$$

The coefficients of the Fourier series are denoted by the following:(21)$$\left\{ \begin{array}{l} \begin{array}{l} {a^{\prime }}_{0}=1/T\int_{-T/2}^{T/2}x(t)dt\\ {a^{\prime }}_{n}=2/T\int_{-T/2}^{T/2}x(t)\cdot \cos n\cdot \tilde{w}tdt\\ {b^{\prime }}_{n}=2/T\int_{-T/2}^{T/2}x(t)\cdot \sin n\cdot \tilde{w}tdt \end{array}\\ \tilde{w}=2\pi /T \end{array}\right.$$
where ${a^{\prime }}_{0}$, ${a^{\prime }}_{n}$, and ${b^{\prime }}_{n}$ are the coefficients to be determined, n is the number of harmonics, and $\tilde{w}$ is the base frequency.

Since Fourier series fitting cannot effectively identify the signal outside the frequency band, in order to fit the target signal outside the frequency band output by the MEMS gyroscope, a linear term $\ell_{n}$ is added on the basis of Fourier sine and cosine components in this paper, and this linear term is used for linear fitting of signals outside the range of $\left(\tilde{w},M\tilde{w}\right)$.

The improved Fourier fitting algorithm is shown in Equation (22).
(22)$$x_{nk}=\{\begin{array}{ll} {a^{\prime }}_{n}\cdot \cos(n\tilde{w}G_{k})+{b^{\prime }}_{n}\cdot \sin(n\tilde{w}G_{k}) & n=1,2,\cdots 2M\\ \ell_{n} & n=2M+1 \end{array}$$
(23)$${℘}_{k}=\omega_{k}-{ℑ}_{k}^{T}X_{k}$$
(24)$${ℑ}_{k+1}={ℑ}_{k}+2\xi {℘}_{k}X_{k}$$
where $X_{k}$ is a column vector with dimension 2M+1, and its specific element $x_{nk}$ is defined by Equation (22), which is used to represent the results of fitting each sine and cosine, ${ℑ}_{k}$ is also a column vector with dimension 2M+1, and each element ${ℑ}_{nk}$ is the weight of the corresponding element $x_{nk}$ in $X_{k}$, $G_{k}$ is the MEMS gyroscope output angular velocity, $\omega_{k}$ is the ideal output value of the MEMS gyroscope, ${℘}_{k}$ represents the error between the estimated value of the gyroscope output and the true value, ξ is the convergence rate coefficient of the algorithm, and its magnitude determines the rate of convergence.

In the process of fitting, the least squares algorithm is used to obtain the optimal weight. From Equation (23) we can obtain:(25)$$\tilde{\varpi }=E({℘}_{k}^{2})=E(\omega_{k}^{2})+{ℑ}^{T}Rℑ-2E(\omega_{k}X_{k}^{T})ℑ$$
where E represents the expected value, $R=E(X_{k}X_{k}^{T})$ is the correlation matrix of the input vector, and ℑ is the weight vector.

The determination of the optimal weights is based on the principle of minimizing the fitting error, and according to the least squares method, the vector of weights is derived as:(26)$$\partial \tilde{\varpi }/\partial ℑ=2Rℑ-2E(\omega_{k}X_{k}^{T})$$

When $\partial \tilde{\varpi }/\partial ℑ=0$, the optimal weight vector can be determined as:(27)$${ℑ}^{*}=R^{-1}\cdot E(\omega_{k}X_{k}^{T})$$

In the process of solving the optimal weight vector by using Equation (27), it is necessary to solve the linear equations. Before solving the equations, it is necessary to invert the correlation matrix, but the calculation of the inverse matrix is large. In order to reduce the amount of calculation, the optimal weight vector uses the steepest descent algorithm.

The steepest descent algorithm is used to find an optimal solution ${ℑ}_{0}$, so that:(28)$$J({ℑ}_{0})\leq J(ℑ)$$

In each iteration, the weights of the input vector satisfy:(29)J(ℑ(m+1))≤J(ℑ(m))
where m is the number of iterations.

In the iteration process, ℑ is continuously adjusted along the inverse gradient of J(ℑ) to obtain:(30)ℏ=∇J(ℑ)=∂J(ℑ)/∂ℑ
where ℏ represents the gradient vector.

From the above, the steepest descent method can be expressed as:(31)ℑ(m+1)=ℑ(m)−νℏ(m)/2
(32)Δℑ(m)=−νℏ(m)/2
where ν is a positive step parameter.

To illustrate the convergence of the steepest descent method, Taylor expansion is performed at ℑ(m):(33)$$J(ℑ(m+1))\approx J(ℑ(m))+\nu^{H}(m)\cdot \Delta ℑ(m)$$

Through the analysis of the steepest descent algorithm, when $0<\nu <2/\lambda_{\max}$, the steepest descent method is convergent, where $\lambda_{\max}$ is the maximum eigenvalue of the correlation matrix R. The fitting result of the MEMS gyroscope output is:(34)$${\hat{\Omega }}_{k}={ℑ}_{1k}\cdot x_{1k}+{ℑ}_{2k}\cdot x_{2k}+\cdots +{ℑ}_{2Mk}\cdot x_{2Mk}+{ℑ}_{(2M+1)k}\cdot \ell_{2M+1}$$

According to Equation (16), the output of the MEMS gyroscope is affected by random error. Therefore, this paper uses the strong noise reduction ability of the Fourier series to correct the residual of the fitting model (Equation (34)) so as to improve the accuracy of the fitting model.

The residual sequence of the fitting model can be expressed as:(35)Ek={e1,⋯,ek,⋯,ej}
(36)$$e_{k}=\omega_{k}-{\hat{\Omega }}_{k}$$

The residual sequence can be approximately expressed by Fourier series expansion as follows:(37)$$e_{k}\approx a_{0}/2+\sum_{i=1}^{k_{a}}\left[a_{i}\cos(2\pi ik/T)+b_{i}\sin(2\pi ik/T)\right]$$
where, $a_{0}$, $a_{i}$, and $b_{i}$ are the coefficients to be determined, $k_{a}=j/2$.

The matrix form of Equation (37) is:(38)Ek≈PaCa
where, Eak=[e1e2⋯ej]T is the residual vector, $C_{a}={\left[\begin{array}{cccccc} a_{0} & a_{1} & b_{1} & \cdots & a_{ka} & b_{ka} \end{array}\right]}^{T}$ is the residual Fourier coefficient vector, and $P_{a}$ can be expressed as:(39)$$P_{a}=\left[\begin{array}{cccccc} 1/2 & \cos(1\times 2\pi \times 1/T) & \sin(1\times 2\pi \times 1/T) & \cdots & \cos(k_{a}\times 2\pi \times 1/T) & \sin(k_{a}\times 2\pi \times 1/T)\\ 1/2 & \cos(1\times 2\pi \times 2/T) & \sin(1\times 2\pi \times 2/T) & \cdots & \cos(k_{a}\times 2\pi \times 2/T) & \sin(k_{a}\times 2\pi \times 2/T)\\ \vdots & \vdots & \vdots & & \vdots & \vdots \\ 1/2 & \cos(1\times 2\pi \times j/T) & \sin(1\times 2\pi \times j/T) & \cdots & \cos(k_{a}\times 2\pi \times j/T) & \sin(k_{a}\times 2\pi \times j/T) \end{array}\right]$$

The coefficient vector obtained by the least squares method is:(40)C^a=(PaTPa)−1PaTEa

Substituting the obtained Fourier coefficient vector ${\hat{C}}_{a}$ into Equation (38), E^k can be obtained, so as to obtain the modified model:(41)Ω^Fk=Ω^k+E^k

## 4. Experiments and Results

### 4.1. Experiment Setup

To verify the effectiveness of the presented method, the compensation process was performed on an AG-20 sensor (Table 1). AG-20 is a three-axis MEMS gyroscope fabricated by polysilicon micromachining. The experimental environment is shown in Figure 4. The experimental equipment includes a biaxial angular velocity turntable, a data acquisition system, and a processing system. The IMU is connected to a laptop computer through an RS485 interface, which is used for information exchange with the computer.

### 4.2. Analysis of Compensation Results for MEMS Gyroscope

The MEMS gyroscope (AG-20) is fixed horizontally on the rate turntable with the tooling fixture, and the Y-axis is selected as the sensitivity axis. Preheating the MEMS gyroscope lasted for 20 min. In the process of experiment, in order to provide a tiny angular velocity for the MEMS gyroscope, a static experiment with a 10-degree interval was carried out on the turntable. The turntable was at an azimuth angle of 0°, 10°, 20°, 30°, 40°, 50°, 60°, 70°, 80°, 90°, 100°, 110°, 120°, 130°, 140°, 150°, 160°, 170°, 180°, 190°, 200°, 220°, 230°, 240°, 250°, 260°, 270°, 280°, 290°, 300°, 330°, 340°, and 350° in order of rotation. The gyroscope output data were collected for three minutes at each azimuth with a sampling frequency of 100 Hz. Therefore, 36 components of the angular velocity of the Earth’s rotation can be obtained as inputs to the MEMS gyroscope. Randomly selecting the original output data of the MEMS gyroscope at four of these azimuth angles, they are shown in Figure 5. In Figure 5, the red line, blue line, pink line, and green line are the original output data of the MEMS gyroscope when the azimuth angles are 10°, 70°, 170°, and 240°, respectively.

When the experiment is completed and all the data are collected, the relationship between the input and output of the MEMS gyroscope at different azimuths is shown in Figure 6. The relative errors of the MEMS gyroscope output at different azimuths are shown in Figure 7.

Figure 6 shows the relationship between the input and output of the MEMS gyroscope at different azimuths. It can be seen that the output of the MEMS gyroscope presents different degrees of nonlinearity with the change in azimuth. The standard value in Figure 6 represents the input angular velocity of the MEMS gyroscope. In this paper, the earth’s rotation angular velocity component is used as the input of the MEMS gyroscope, so the standard value can be calculated by Equation (15). The experimental data in Figure 6 are averaged by collecting three-minute output data from the MEMS gyroscope. As seen in Figure 7, when the azimuth angle is close to 90° or 270°, the relative error is the largest, and it increased to 85.13% and 81.42%, respectively. The reason for this is that the angular velocity component of the Earth’s rotation is close to zero, and the MEMS gyroscope that acquires the effective signal is so tiny that the nonlinearity is more noticeable.

In this paper, 20 data points were randomly selected to establish the model, and the remaining 16 data points were utilized to validate the model and compare the compensation effects of different compensation methods. Figure 8 shows the fitting results of different compensation methods. The fitting residuals of different compensation methods are shown in Figure 9.

Figure 8 shows the fitting effect of different compensation methods. The fitting effect can be judged by the coefficient of determination (R-square), and the closer the R-square is to 1, the better the fitting effect. The R-square of polynomial fitting is 0.8203, the R-square of the Fourier series is 0.9056, and the R-square of AFCM is 0.9996. It can be seen from Figure 9 that the residuals with AFCM are smaller and more concentrated than those of polynomial fitting and Fourier series fitting. Therefore, the fitting effect of AFCM is the best.

The remaining 16 data points were used to validate models fitted by different compensation methods. The results are shown in Figure 10, with the specific values given in Table 2.

The output results of the MEMS gyroscope after compensation by different methods are displayed in Figure 10. The red line in Figure 10 is the original output of the MEMS gyroscope, which has a large error compared with the standard value (purple line). After compensation by the traditional polynomial fitting method (cyan line) and the Fourier series fitting method (blue line), the output error of the MEMS gyroscope is reduced to varying degrees, but there are still some errors. The output of the MEMS gyroscope is closer to the standard value after AFCM compensation (green line). In Table 2, we summarize the absolute error, mean error, and standard deviation (STD) of MEMS gyroscope output after different compensation methods. We evaluate the results with the standard deviation. The proposed compensation method reduces the STD by 96.15%, 77.92%, and 42.47% compared to the raw data, traditional polynomial, and Fourier series, respectively. The calculation method is shown in Equations (42)–(44). The results show that the output error of the MEMS gyroscope is the smallest and the stability is the best after AFCM compensation.
(42)$$\eta_{raw}^{AFCM}=\frac{STD_{raw}-STD_{AFCM}}{STD_{raw}}*100\%=\frac{0.000815143-3.1414\times 10^{-5}}{0.000815143}*100\%=96.15\%$$
(43)$$\eta_{polynomial}^{AFCM}=\frac{STD_{polynomial}-STD_{AFCM}}{STD_{polynomial}}*100\%=\frac{0.000142305-3.1414\times 10^{-5}}{0.000142305}*100\%=77.92\%$$
(44)$$\eta_{fourier}^{AFCM}=\frac{STD_{fourier}-STD_{AFCM}}{STD_{fourier}}*100\%=\frac{5.4609\times 10^{-5}-3.1414\times 10^{-5}}{5.4609\times 10^{-5}}*100\%=42.47\%$$
where $\eta_{raw}^{AFCM}$, $\eta_{polynomial}^{AFCM}$, $\eta_{fourier}^{AFCM}$ are the percentages of STD reduction after compensation by the proposed method, $STD_{raw}$ is the STD of the MEMS gyroscope’s raw output error, $STD_{polynomial}$ is the STD of MEMS gyroscope output error after polynomial compensation, $STD_{fourier}$ is the STD of MEMS gyroscope output error after Fourier series compensation, and $STD_{AFCM}$ is the STD of MEMS gyroscope output error after AFCM compensation.

Figure 11 shows the nonlinearity before and after compensation by different methods. As we can see, the MEMS gyroscope exhibits significant nonlinearity (the red line) at a tiny angular velocity. After compensation by the traditional polynomial method (the cyan line) and the Fourier series fitting method (the blue line), the nonlinearity is reduced, but there is still some nonlinearity. After AFCM (green line) compensation, the nonlinearity of the MEMS gyroscope is greatly reduced. It can be seen from Figure 10 that the raw nonlinearity of the MEMS gyroscope output is 1150.87 ppm. After compensation by the traditional polynomial method, the nonlinearity is 641.13 ppm. After compensation by the Fourier series method, the nonlinearity is 250.55 ppm. After compensation by the AFCM, the nonlinearity is reduced to 68.89 ppm. The compensation method proposed in this paper substantially improves the output accuracy of the MEMS gyroscope, and the accuracy consistency of the whole range is better.

## 5. Conclusions

In this paper, the output precision of the MEMS gyroscope is improved by compensating for the nonlinearity of the MEMS gyroscope at tiny angular velocities. The key contributions of this paper are to improve the Fourier series fitting method, then use the steepest descent method to solve the optimal weight, and finally, use the Fourier series to correct the residual error of the fitted model. Based on the experimental results and analyses, the following conclusions can be drawn: (a) by comparing the results of traditional polynomial fitting and Fourier fitting, the evaluation standard of fitting residuals (R-square) is obtained, and the results show that the proposed method has a better fitting effect; (b) the availability and effectiveness of the AFCM are demonstrated by an obvious decrease in the standard deviation of the MEME gyroscope output error after compensation, which is shown in Table 2; (c) the output nonlinearity of the MEMS gyroscope is reduced from 1150.87 ppm to 68.89 ppm, and the compensation effect is better than that of the traditional polynomial method and the Fourier series method. The proposed method can effectively compensate for the nonlinearity of the MEMS gyroscope under tiny angular velocity and exhibits wide application prospects.

## Figures and Tables

**Figure 1 sensors-22-06577-f001:**
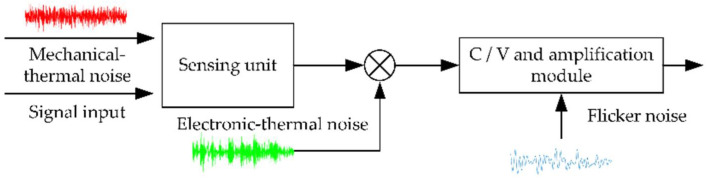
The output of MEMS gyroscope affected by noise.

**Figure 2 sensors-22-06577-f002:**
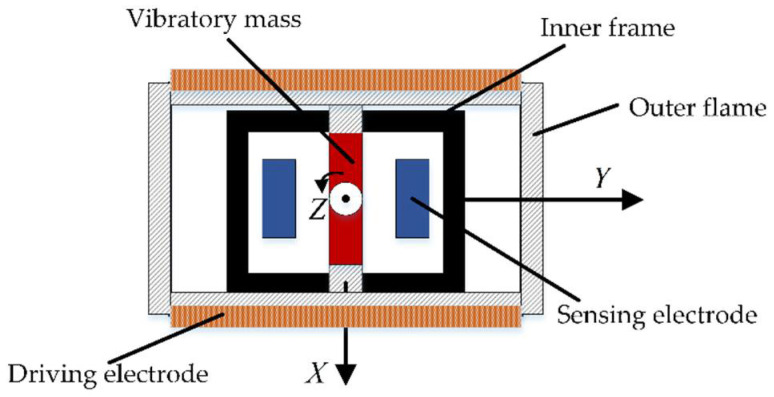
MEMS gyroscope structure diagram.

**Figure 3 sensors-22-06577-f003:**
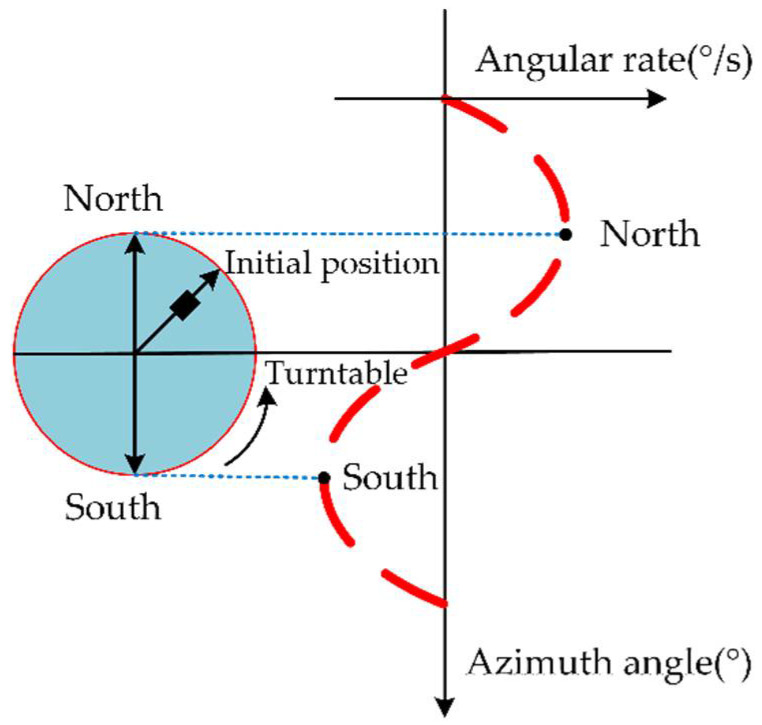
Curve of earth and turntable combined action.

**Figure 4 sensors-22-06577-f004:**
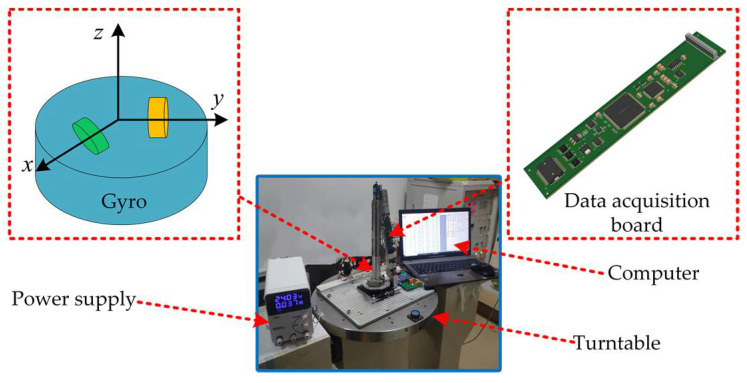
Experimental equipment.

**Figure 5 sensors-22-06577-f005:**
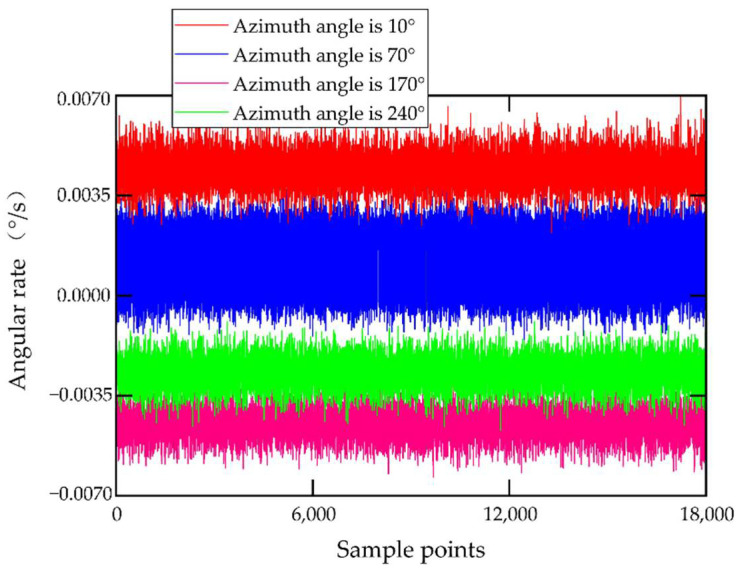
Original data of the MEMS gyroscope.

**Figure 6 sensors-22-06577-f006:**
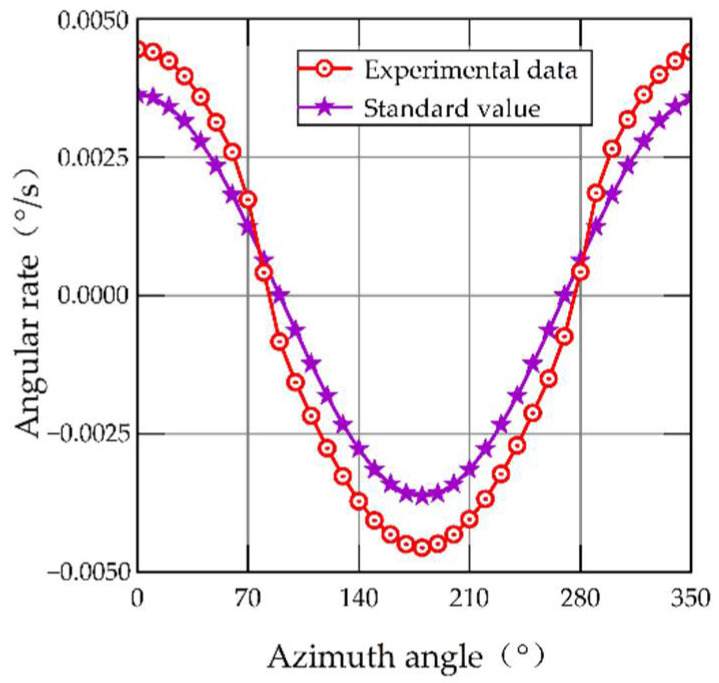
MEMS gyroscope output at different azimuths.

**Figure 7 sensors-22-06577-f007:**
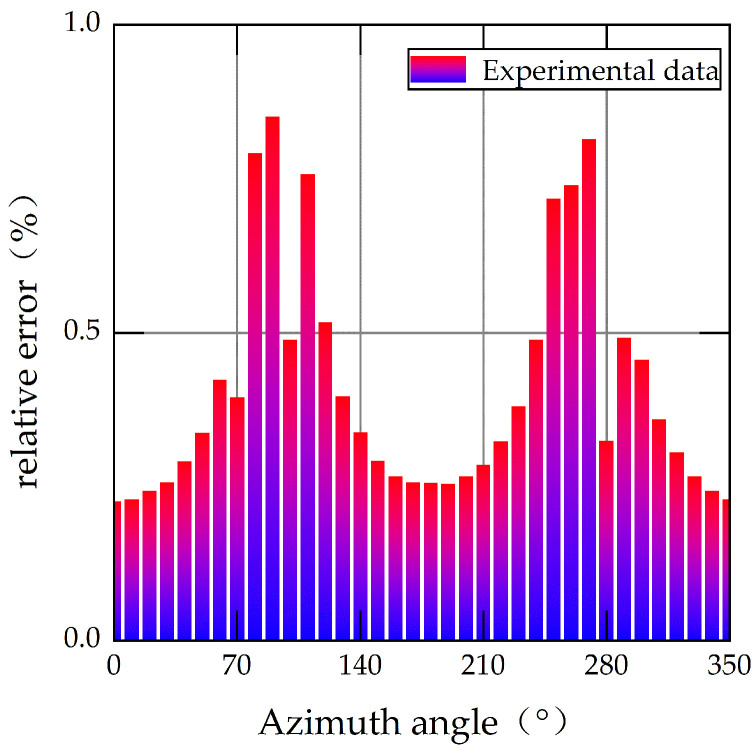
Relative error of MEMS gyroscope output.

**Figure 8 sensors-22-06577-f008:**
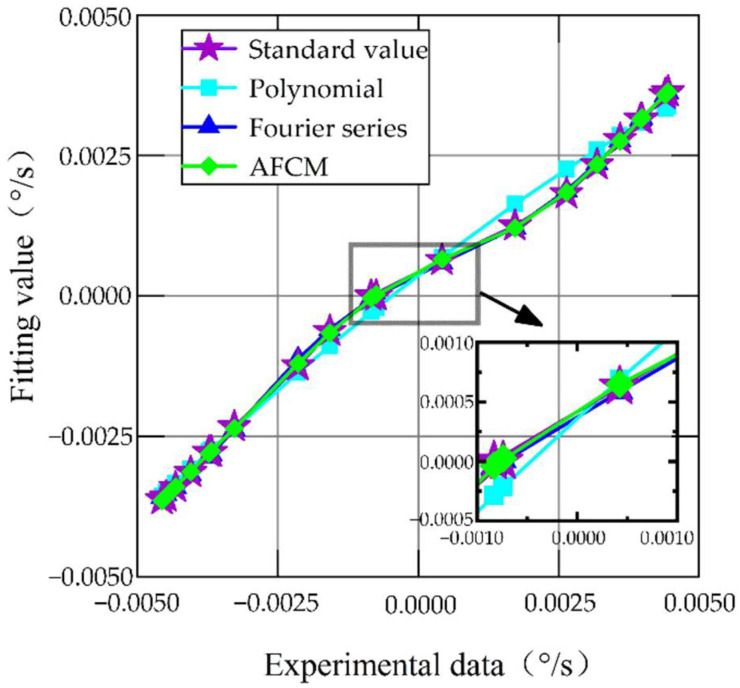
Fitting results of different compensation methods.

**Figure 9 sensors-22-06577-f009:**
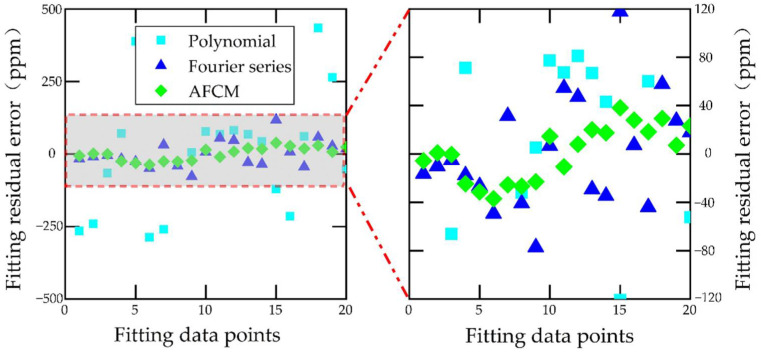
Fitting residuals of different compensation methods.

**Figure 10 sensors-22-06577-f010:**
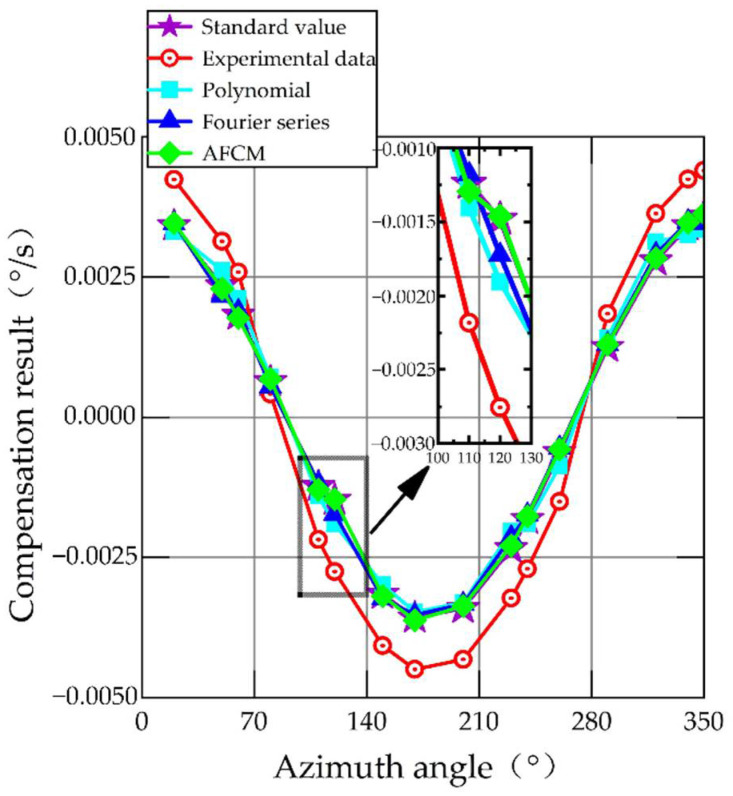
Output results of the MEMS gyroscope with different compensation methods.

**Figure 11 sensors-22-06577-f011:**
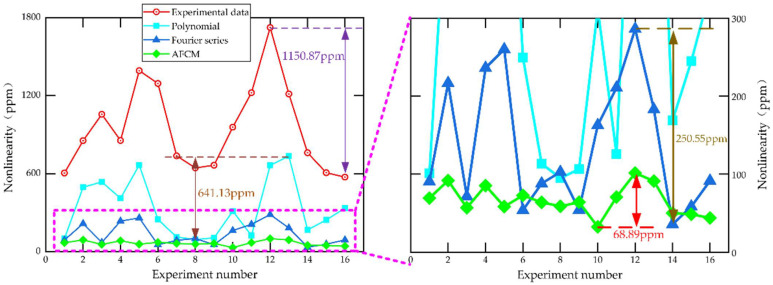
Comparison of nonlinear error compensation results.

**Table 1 sensors-22-06577-t001:** AG-20 Specifications.

**Working Temperature**	**S** **upply V** **oltage**	**W** **orking C** **urrent**	**B** **andwidth**
−40 °C~+150 °C	5 V ± 0.1 V (DC)	≤300 mA	≥12 Hz
**Measuring Range**	**Zero-Bias Stability**	**Bias Repeatability**	**Resolution**
±100°	≤0.3°/h	≤0.5°/h	≤0.1°/h

**Table 2 sensors-22-06577-t002:** Comparison table of compensation method error results.

Azimuth Angle (°)	Raw Data (°/s)	Polynomial (°/s)	Fourier Series (°/s)	AFCM (°/s)
20	0.000826439	−6.89415 × 10^−5^	6.1758 × 10^−5^	4.72656 × 10^−5^
50	0.000798303	0.000231583	−0.000101278	−4.28319 × 10^−5^
60	0.00076815	0.000194944	2.59435 × 10^−5^	−2.06786 × 10^−5^
80	−0.000215985	5.19142 × 10^−5^	−2.97954 × 10^−5^	1.07552 × 10^−5^
110	−0.000692641	−0.000165493	6.46396 × 10^−5^	−1.45021 × 10^−5^
120	−0.00094015	−9.06175 × 10^−5^	1.95771 × 10^−5^	−2.64337 × 10^−5^
150	−0.000927123	7.13357 × 10^−5^	^−5^.54034 × 10^−5^	−4.01626 × 10^−5^
170	−0.000921504	0.000067901	7.34738 × 10^−5^	−4.20862 × 10^−5^
200	−0.000908439	7.25774 × 10^−5^	3.69618 × 10^−5^	4.3723 × 10^−5^
230	−0.000894303	0.000145932	7.60111 × 10^−5^	1.51402 × 10^−5^
240	−0.00088915	−4.56297 × 10^−5^	7.65545 × 10^−5^	2.56077 × 10^−5^
260	−0.000435326	−8.371 × 10^−5^	3.61233 × 10^−5^	1.27842 × 10^−5^
290	0.000603201	0.00018294	4.55104 × 10^−5^	2.26422 × 10^−5^
320	0.000846424	9.39976 × 10^−5^	1.98744 × 10^−5^	2.78241 × 10^−5^
340	0.000829439	−0.000167173	3.98373 × 10^−5^	3.31562 × 10^−5^
350	0.000821504	−0.000239669	−6.54447 × 10^−5^	3.14372 × 10^−5^
Average (°/s)	−8.31975 × 10^−5^	1.57433 × 10^−5^	2.02715 × 10^−5^	5.22753 × 10^−6^
STD (°/s)	0.000815143	0.000142305	5.4609 × 10^−5^	3.1414 × 10^−5^

## Data Availability

Not applicable.

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
