# Peer review of "Research on Nonlinear Compensation of the MEMS Gyroscope under Tiny Angular Velocity"

_sensors, 2022, doi:10.3390/s22176577_

Round 1
Reviewer 1 Report
Dear Sir,
I have reviewed the paper and read it multiple times and tried my level best to understand the mathematics involved in it. Below are my some concerns and observations to the best of my knowledge.
Overall, the work is good, especially, in terms of model development of AFCM.
COMMENTS
Section 1
The intention of the study is clearly spelled out with reasonable justification. The authors spelled out the urgency of such study. Several feedbacks are offered to improve the introduction.
However, the explanation of research gap is limited and not sufficient. Authors have highlighted different compensation methods for high angular velocity application but no literature review has been described related to the compensation of nonlinear errors in tiny angular velocity for MEMS gyroscope. Is this the first ever work on tiny angular velocity compensation for MEMS gyroscope? If that is the case, then it should be stated in the introduction. If not, then the work done already in tiny angular velocity error compensation should be highlighted in the introduction with proper referencing. Why tiny angular velocity error compensation should be the focus in this paper? A few lines on the benefits of this work should be added to further explain the importance of the work.
Section 2
In addition to the tiny angular velocity errors, authors have highlighted different other errors that effect the output of MEMS gyroscope. These errors include random and deterministic error and errors due to geometric and material effects, electrostatic actuation, capacitance detection, and other factors.
But readers are more interested to know which error (s) are the point of focus in this work? Is only the error due to tiny angular been added into the output model of MEMS gyroscope? Have the authors included the contributions of all the other errors/variables in the output model of MEMS gyroscope presented in section 2?
Section 3
Authors have mainly compared the adaptive Fourier series compensation method AFCM performance with the traditional polynomial compensation method. However, authors should also discuss and argue more in detail the rationale behind choosing the adaptive Fourier series compensation method (AFCM) among all the iterative methods. What ae the limitations of the AFCM especially in MEMS gyroscope output error modelling?
The model of AFCM in section 3 does entail all the required steps for the residual correction and is clearly defined.
Section 4
Out of 36 azimuths points, authors used 20 data points to establish model and remaining 16data points to validate model and compare the compensation effects of different compensation methods. AFCM R- square value of 0.9996 and the presented results in Figure 5 to Figure10 and Table 2 does validate the performance of AFCM. Hence, the theoretical, and practical implication is adequately discussed.
A more elaborative conclusion is required to further strengthen author argument and its relation to the introduction should be established.
Proofreading is needed for this manuscript to improve the readability.
Author Response
Dear Reviewer,
Your advice and questions are greatly appreciated. We carefully checked all your comments and revised our manuscript accordingly. The changes that have been made are shown in the revised manuscript in a yellow highlight. Meanwhile, references and typo errors have been checked and corrected.
Please see the attachment.

Reviewer 2 Report
1. The manuscript will benefit from thorough proofreading. There is significant scope for improvement to benefit the readers.
2. Pls rewrite the abstract
3. Page 3, “Poly-silicon or crystal silicon are the primary materials used in the MEMS gyroscope. As a result, the MEMS gyroscope output is affected by micro-size effects, manufacturing processes, and various noises, which are the main reasons for the nonlinearity of the MEMS gyroscope. – This sentence and the usage of “As a result” are unclear to me. What are the micro-scale effects, and what is the meaning of “various noises”? How is the manufacturing process affect the nonlinearity of the gyroscope? Kindly clarify.
4. Explain figure 2. The working principle of Gyro considered in the present study with close-in views of various components needs to be elaborated.
5. Explain the physical meaning and significance of the equations included under various models.
6. Pls, comment on whether the considered AG-20 is a micro-gyroscope made of single crystal silicon or poly-silicon using micro-fabrication techniques.
7. What is the meaning of the standard value in Figure 5? Pls, comment on whether the data plotted in Figure 6 is repeatable. Why is the relative error positive for three discrete angles and negative for all other angles?
8. As seen in Figure 6, when the azimuth angle is 90 degrees or 270 degrees, the relative error increases to 85.13 percent. – 85.13? Pls, cross-check.
9. Pls, explain how the percentage values were calculated in the following statement. “ The proposed compensation method reduces the STD by 96.15%, 77.92%, and 42.47% compared to the raw data, traditional polynomial, and Fourier series, respectively.” I would appreciate one example calculation.
Author Response

(The authors gave the same response as above.)

Reviewer 3 Report
This variant of the manuscript has to be completely revised. It contains some interesting results, which are worth publishing in “Sensors”, but this text is to be thrown away and a brand new manuscript has to be furnished.
First of all, the authors are to find somebody with the better English. The current version contains numerous sentences which we cannot even understand. Just two examples from the second page of the manuscript. In the second paragraph there is the statement “However, polynomial fitting is difficult to accurately describe the output characteristics of the MEMS gyroscope.” In the next paragraph one more strange statement “The gyroscope outputs the equivalent angular velocity signal by applying the voltage signal, and then compensates the MEMS gyroscope nonlinearity according to the output signal [22,23].”. We simply do not understand these two and many other statements further on.
The authors are to be careful with the term “nonlinearity”. In physics and in engineering this term is reserved for the situations when some dependence is linear until the parameter reaches higher values. Such situation is discussed in the second page of the manuscript. However, the case under consideration deals with low values. To our opinion, in this case we cannot speak about the nonlinearity, but only about nonlinear behavior or, better, deviation from the linear dependence.
The physical nature of such deviation is not considered in the manuscript. The paper contains only some strange Figure 1, given practically without explanations, and then authors simply cite some mathematical relationships from other papers without explanation of their physical sense.
The authors are also to describe their method not only as some mathematics, but also in the technical terms, which are to be understandable by the practical engineers. In the manuscript they do not explain the idea of compensation of the nonlinear behavior. They first briefly describe the polynomial fitting – again, only as citations from other papers, without proper consideration. Then they transit to the Fourier fitting. They say that the subject of the paper is the adaptive variant of such fitting, but in fact we do not find neither explanation of the “usual” (non-adaptive) approach nor references to some earlier papers.
Again, the explanation of the method is given only in mathematical way – without technical details. We understand how the Fourier series fit works when the gyroscope is subjected to the cyclic variation of the angular velocity, but we have simply failed to understand the transfer of the method application from cyclic excitation to the measurements of the unknown angular velocity.
The experiment has also to be described in much more details. We are to know the gyroscope parameters, to know the measurement procedure and to see some raw data.
Author Response

(The authors gave the same response as above.)

Round 2
Reviewer 2 Report
The authors have addressed most of my comments satisfactorily. A few minor optional suggestions are given hereunder:
1) Pls look into the title. The word research may be dropped. A suggestion is"Nonlinear Compensation of a Polysilicon MEMS Gyroscope under Tiny Angular Velocities". Anyhow, I leave it to the authors to decide.
2) Is it possible to provide a close-in view of the gyroscope in Fig. 4?
Reviewer 3 Report
We are satisfied by corrections and think now that the paper is ready for publication.